# Evidence of an Absence of Inbreeding Depression in a Wild Population of Weddell Seals (*Leptonychotes weddellii*)

**DOI:** 10.3390/e25030403

**Published:** 2023-02-22

**Authors:** John H. Powell, Steven T. Kalinowski, Mark L. Taper, Jay J. Rotella, Corey S. Davis, Robert A. Garrott

**Affiliations:** 1Department of Ecology, Montana State University, P.O. Box 173460, Bozeman, MT 59717, USA; 2Department of Biological Sciences, University of Alberta, Edmonton, AB T6G 2E9, Canada

**Keywords:** data cloning, microsatellites, individual inbreeding coefficient, lifetime reproductive success, statistical evidence, information criteria

## Abstract

Inbreeding depression can reduce the viability of wild populations. Detecting inbreeding depression in the wild is difficult; developing accurate estimates of inbreeding can be time and labor intensive. In this study, we used a two-step modeling procedure to incorporate uncertainty inherent in estimating individual inbreeding coefficients from multilocus genotypes into estimates of inbreeding depression in a population of Weddell seals (*Leptonychotes weddellii*). The two-step modeling procedure presented in this paper provides a method for estimating the magnitude of a known source of error, which is assumed absent in classic regression models, and incorporating this error into inferences about inbreeding depression. The method is essentially an errors-in-variables regression with non-normal errors in both the dependent and independent variables. These models, therefore, allow for a better evaluation of the uncertainty surrounding the biological importance of inbreeding depression in non-pedigreed wild populations. For this study we genotyped 154 adult female seals from the population in Erebus Bay, Antarctica, at 29 microsatellite loci, 12 of which are novel. We used a statistical evidence approach to inference rather than hypothesis testing because the discovery of both low and high levels of inbreeding are of scientific interest. We found evidence for an *absence* of inbreeding depression in lifetime reproductive success, adult survival, age at maturity, and the reproductive interval of female seals in this population.

## 1. Introduction

In his seminal work *The Logic of Scientific Discovery*, the philosopher Karl Popper argued that the scientific method should be based on the falsification of hypotheses [1]. Modern statistical approaches provide powerful tools for this approach to scientific inquiry but fail to provide evidence for a null hypothesis if it is not refuted [2]. This failure to provide evidence of the null is problematic because well-constructed null hypotheses are meaningful. The inability of modern statistical approaches to hypothesis testing to address the validity of the null hypothesis can lead to a misinterpretation of the results and/or a reduced likelihood of publication [3], a phenomenon known as the file-drawer problem [4,5].

Inbreeding depression is a field of research that could be negatively affected by current approaches to statistical hypothesis testing because the null hypothesis for these studies, that inbreeding depression is not occurring in a population, is of interest in the conservation and management of species. The reduction in fitness of inbred individuals is one of the oldest observations of population genetics [6]. Inbreeding depression was first recognized among domesticated species [7], but it also affects captive and wild populations [8,9,10,11,12]. The negative effects of inbreeding are usually stronger in the wild [10] and have the potential to reduce the viability of these populations [12,13]. Scientists are often tempted to use the failure to reject the null hypothesis as evidence that the null hypothesis is true. This interpretation is number two on Goodman’s list of common misinterpretations of *p*-values [14]. This move is illegitimate and sometimes misleading under classical statistical testing procedures, which is demonstrated by [15] in the chapters on Neyman–Pearson hypothesis testing and Fisherian null hypothesis significance testing. This desirable inferential move is accessible to scientists by using an evidential statistics approach. The evidentialist is able to say either (1) the “null” model is better supported than the alternative, if an explicit alternative is specified; or (2) the support for the alternative over the null is no greater than the expected bias due to the estimation of unspecified parameters [2,16]. We want to emphasize that the evidentialist is not committing the fallacy of argument from ignorance [17]. Evidence is a data-based estimate of the contrast of the divergence of the distributions implicit in each of the compared models to the distribution of the observations [2,16,18,19].

Although there are many methods available to test for inbreeding depression (reviewed in [11]), they are all based on a comparison of fitness traits between inbred and non-inbred individuals. Whether inbreeding depression is estimated using a regression model assuming additive effects of loci [11] or multiplicative effects of loci [20], inbreeding coefficients are included as explanatory variables and are assumed to be known quantities. Calculating inbreeding coefficients from pedigrees has been the gold standard in estimating these parameters [21]. Although measurement error can exist in pedigree inbreeding coefficients due to errors in the pedigree [22], non-zero inbreeding coefficients of the founders [22,23], or relatedness of the founders [22], a high correlation in individual inbreeding coefficients calculated from shallow (4 to 5 generations) and deep (50 generations) pedigrees [23] indicates that pedigree inbreeding coefficients are effective estimators of inbreeding.

Multilocus heterozygosity has been used as a surrogate for pedigree inbreeding coefficients in studies of inbreeding depression when pedigree-based estimates are not available because of the established relationship between genome-wide heterozygosity and inbreeding [24]. Although substituting multilocus heterozygosity for pedigree inbreeding coefficients avoids potentially difficult or impossible historic data collection, empirical [22,25] and simulation [23] studies indicate that multilocus heterozygosity is only weakly associated with individual inbreeding coefficients. In addition, the effect size of these studies is often small, and the average number of analyzed loci is well below the predicted number needed to explore these patterns [26]. Recent advances in genotyping technologies have increased the availability of, and reduced the cost to generate, individual genotypes at the genomic scale (e.g., [27]), and estimates of inbreeding coefficients from genomic data have been shown to outperform pedigree estimates of inbreeding [28].

However, what are we to make of studies completed prior to the widespread application of these next-generation sequencing technologies? The potential exists for earlier studies of heterozygosity fitness correlations to be affected by a publication bias, although meta-analyses disagree on this occurring (e.g., [29,30] found evidence for a bias, but [26] did not). Here we present a study conducted as a portion of the lead author’s dissertation research [31] that failed to statistically reject the null hypothesis that inbreeding depression did not reduce the lifetime reproductive success of a population of Weddell seals (*Leptonychotes weddellii*). This result reduced interest in pursuing the subsequent publication of this work, making it at an example of the file-drawer effect.

We present the study here largely as it was developed for initial publication, with added discussion of how this paper and its publication history relate to evidential statistics and the file-drawer effect. In this study, we completed a two-step maximum likelihood analysis, which was fitted using the data-cloning algorithm [32], and it incorporated the error in estimating inbreeding coefficients from multilocus genotypes into estimates of inbreeding depression. We took an information-criterion-based model-selection approach to compare models estimating the association between fitness parameters and estimated inbreeding coefficients in a Weddell seal population that has been the focus of a long-term ecological study in Erebus Bay, Antarctica. Information criteria, and evidential statistics in general, treat all models equally; in effect, there is no null hypothesis, only multiple alternatives.

## 2. Materials and Methods

### 2.1. Study Area and Organism

Weddell seals in Erebus Bay, Antarctica, have been the focus of a long-term ecological study since 1969 [33,34]. In this population, between 8 and 14 colonies of female Weddell seals form each austral spring along tidal cracks in the sea ice (see [35] for a map) to pup [34]. Individuals in this population have been uniquely marked and annually surveyed [33,34], and, as such, there is a wealth of information available about the survival and reproductive output of female Weddell seals. Females in this population reach maturity between 4 and 14 years of age [36,37] and produce a single pup [38] every 1.5–2.2 years [39]. Individual seals exhibit strong philopatry [35], resulting in most reproductive events of mature females being known. Within Erebus Bay, individual survival and age at first reproduction are known to be associated with the breeding colony in which an individual was born [37,40], as well as the year of an individual’s birth [41]. In addition, this population of seals has founded a smaller isolate on White Island, Antarctica, in which inbreeding is associated with reduced pup survival [42].

### 2.2. Sample Selection and Genotyping

One hundred and fifty-four individual female seals, born between 1980 and 1992, were genotyped. Seal lifetime ranged from a minimum of 7 years to an individual who was observed in the population for a total of 29 years. Because the most recent year class of sampled individuals had a maximum 19-year history in the population, all life histories were truncated to a maximum of 19 years. By tracking reproductive individuals for 19 years, we were able to capture the bulk of the reproductive output of each female because only 9% of all pups in this population were produced by mothers greater than 17 years of age [43]. Tissue samples were collected and stored as described in [42], and individuals were genotyped at 29 microsatellite loci.

Twelve novel microsatellite sequences were isolated by ecogenics GmbH (Switzerland), using the high-throughput genomic sequencing approach of [44] (Table 1). A total of 1 μg of genomic DNA was analyzed on a Roche 454 GS-FLX platform (Roche, Switzerland), using a 1/16th run and the GS FLX titanium reagents. This produced a total of 8001 reads with an average length of 193 bp. Of these, 273 contained a microsatellite insert suitable for primer design with a tetra- or trinucleotide sequence repeated at least six times, or a dinucleotide sequence of at least 10 repeats. Primers were designed for a total of 32 microsatellite inserts, all of which were tested for polymorphism.

The 12 polymorphic microsatellite markers produced by this screen (Table 1) were split into three multiplex reactions, each amplifying four loci. The forward primer of each locus was labeled at the 5′ end with a specific dye (6FAM, VIC, NED and PET, Life Technologies, Carlsbad, CA, USA). Loci in these multiplexes were (1) *Lew-001339*, *Lew-001859*, *Lew-001845*, and *Lew-001677*; (2) *Lew-001873*, *Lew-002658*, *Lew-004467*, and *Lew-002762*; and (3) *Lew-007425*, *Lew-006657*, *Lew-006174*, and *Lew-005761*. Multiplex PCR reactions consisted of 1 µM of each primer, 2 µL 5X MyTaq reaction buffer (Bioline, Taunton, MA, USA), 1 µM M13 primer [45], 0.5 µg MyTaq HS DNA polymerase (Bioline, MA, USA), 1 μL of template DNA, and enough water for a final reaction volume of 10 µL. The PCR profile consisted of one activation step at 95 °C for 15 min, followed by 30 cycles (95 °C for 15 s, 55 °C for 15 s, and 72 °C for 10 s), 10 cycles (95 °C for 15s, 53 °C for 15 s, and 72 °C for 10 s), and a final extension step at 72 °C for 30 min. Microsatellite fragments were visualized using a 3100-Avant Genetic Analyzer, with allele calls made using Genemapper v. 3.7 (Life Technologies, CA, USA). Seals were also genotyped at *Lw-4*, *Lw-7*, *Lw-10*, *Lw-11*, *Lw-16*, *Lw-20*, *Lc-6*, *Lc-13*, *Lc-18*, *Lc-26*, *Lc-28*, *Hi-8*, *Hi-14*, *Hi-15*, *Hi-16*, *Hi-20* [46], and *G1A* [47], as described in [46].

We calculated expected heterozygosity and the mean number of alleles in this sample, using GenAlEx version 6.501 [48,49]. We tested for the presence of pairwise linkage disequilibrium and for whether loci were in Hardy–Weinberg equilibrium, using the exact test of [50], using Genepop on the web [51,52]. For both of these analyses, Markov chains were run for 10,000 iterations in each of the 200 batches.

### 2.3. Two-Step Maximum Likelihood Analysis

The models considered are necessarily hierarchical models (HMs) due to the presence of measurement error. Estimation of and inference on the parameters by classical frequentist methods is difficult, if not impossible, for many HMs because of the high dimension integrals involved (see [32] for discussion). Such models can be estimated in a Bayesian framework, using Markov Chain Monte Carlo (MCMC) simulation. However, both estimation and inference in a Bayesian framework depend on the details of prior and variable transformation used even if priors are deemed noninformative [53]. Data Cloning [32,54,55,56,57] is a powerful algorithmic method for converting *any* Bayesian estimation and inference to a frequentist form free of the influence of prior and transform specification. It is well-known that as sample size increases the posterior distribution in a Bayesian analysis converges to the large sample multivariate normal distribution of the frequentist maximum likelihood estimates [58]. Data cloning tricks a Bayesian algorithm into providing maximum likelihood estimates by providing the algorithm with a series of datasets, each composed of a number of copies (clones) of the original dataset. The number of copies is increased until parameter estimates are stable. Data cloning can also be used to estimate the likelihood ratios needed to calculate the differences of information criteria [56].

Parameter estimates and associated estimates of uncertainty for all analyses in this paper were made using the data-cloning algorithm [32]. This method allows frequentist-based inference to be made from hierarchical models [32] and, as such, makes the hierarchical framework available for any researcher regardless of their data-analysis philosophy. Because the data-cloning algorithm was used to fit the regression models in this study, and cloning the data involved copying both the number of individuals, as well as the genotypic information for each individual, models for estimating individual inbreeding coefficients were run separately from the regression analyses. Due to the additional steps necessary to run a data-cloning analysis, we present its use here. Other researchers interested in a Bayesian analysis can combine the two steps that we present into a single model with the same formulations and make posterior inference from the original data.

### 2.4. Estimating Individual Inbreeding Coefficients

We used the Gibbs sampling method of [59] to estimate individual inbreeding coefficients. This method relates frequencies of the *m^th^* allele at the *l^th^* locus (*p_lm_*) and the inbreeding coefficient of the *i^th^* individual (*F_i_*), which is expressed as a proportion between zero and one, using variables indicating whether or not the alleles at a specific locus (*x_il_*) came from a common ancestor (i.e., were identical by descent). To prevent WinBUGS version 1.4.3 [60] from crashing, we constrained the probability of success for this Bernoulli distribution between 0.00001 and 0.99999 [61].

We assumed that loci were independent in this analysis. While the authors of [59] suggest using a = b = 0.001 for the beta distribution for *F_i_*, we selected the Bayes–Laplace prior (a = b = 1) in order to avoid the unintended informative nature of other common “noninformative” beta prior distributions, such as the Jeffreys prior (a = b = ½) or the Haldane prior (a = b = 0), in the case where individuals have no alleles identical by descent [62]. Data cloning would have eventually overcome the effect of these priors, but the Bayes–Laplace prior speeds the process.

Finally, the vector of indicator variables identifying if alleles were identical by descent was used to specify a vector for each locus in the population (*Z_l_*) that was the total number of alleles identical in appearance (i.e., identical by state) that were not also identical by descent. This was simply the sum of all alleles of a given state across all individuals at the locus, where alleles identical by decent within an individual were counted only once. This vector was used to model the conditional distribution of allele frequencies given identity by descent with a uniform Dirichlet prior distribution [63] on the allele frequencies (*p_lm_*; see [59]).

See [59] for a more detailed description of the Gibbs sampling algorithm used for estimating individual inbreeding coefficient.

The parameters for the beta distribution modeling individual inbreeding coefficients were estimated by fitting the model of [59] to the original data, as well as to datasets that had a total of 5, 10, 20, and 40 copies of the original data, using WinBUGS version 1.4.3 [60] and the dclone package [57] in R version 2.15.0 [64]. All parameter values converged to stable values by a data clone with 40 copies of the data.

Each model was fitted using 3 independent chains run for 10,000 iterations. We saved every 5th sample after discarding the initial 5000 samples. Model convergence to a stationary distribution was assessed by visual inspection of the trace plots and calculating the $\hat{R}$ statistic [65], using the coda package in R [66]. Modal values for the number of loci identical by decent ($\sum_{l}x_{il}$) in the *i^th^* individual were used to parameterize the beta distribution of individual inbreeding coefficients in all cloned regression models. In cases where the posterior distribution was multimodal, the average of the observed modes was calculated and used.

We calculated the correlation between observed individual homozygosity and estimated inbreeding coefficients. We also estimated the identify disequilibrium ($g_{2}$ [67]) using the inbreedR package in R [68].

### 2.5. Modeling the Association between Inbreeding and Fitness

We statistically probed for the presence of inbreeding depression in mature seals in this population by selecting a fitness surrogate that incorporated survival across the 19-year duration of the study, age at first maturity, and the frequency with which a seal reproduced. This statistic is the first eigenvalue of a Leslie matrix constructed for each individual. This value is an individual-based analog (*r_i_*) to the instantaneous population growth rate(*r*). The only difference in the calculation of *r_i_* and *r* is that, in the individual calculation, we included both male and female offspring, while the population calculation considers only female offspring. Our rationale here is that both male and female offspring contribute to an individual’s fitness, but the population’s growth rate is limited by the number of female offspring.

In estimating the relationship between inbreeding and *r_i_*, we included explanatory variables for the birth colony and year of birth for each seal due to their association with survival or reproduction in previous studies [37,40,41]. Birth colony was included as an indicator variable specifying whether or not a seal had been born in one of the Dellbridge Islands colonies (see [37] for a delineation of colonies).

In addition to estimating the relationship between a composite measure of lifetime reproductive success and inbreeding, we also estimated the association between inbreeding and survival across the 19-year duration of the study, mean age at first reproduction, and the frequency of reproductive events. These three additional regression models were run in an effort to identify whether a reduction in fitness at a specific stage of an individual’s life was driving the possible inbreeding depression in lifetime reproductive success.

For estimating the association between individual inbreeding coefficients and *r_i_*, probability of survival, age at first reproduction, and probability of reproducing in a given year, we used BIC and AICc [56,69] to select among up to 10 nested candidate models. For simplicity of presentation in the description of the analyses below, we present the richest model that was fit for each fitness variable, with all other subsets of variables being investigated, as well. Because of known differences in the wait time following first reproduction [37], the probability that an individual reproduced in a given year was modeled with a parallel-lines model differentiating between the wait time following an individual’s first pup and the wait time for all subsequent pups. This model was specified using a variable whose value was one if the interval was following an individual’s first pup and zero otherwise.

Model selection was performed in an effort to reduce the potential loss of precision in estimating the regression coefficient for individual inbreeding coefficient that may occur if too many explanatory variables are in the model [70]. We computed both the BIC [71] (also known as the Schwartz Information Criterion, or SIC) and AICc for model selection. The BIC tends to slightly underfit (see [2,72] for discussion), and the AIC family of selection criteria (AIC, AICc, and TIC) is slightly biased toward selecting more complex models. The total number of parameters was assumed to equal the number of regression parameters (k = 1, …, 4). Because *r_i_* was modeled as a normally distributed variable, an additional variance parameter was also included in calculating the total number of parameters for the models estimating the relationship between *r_i_* and the inbreeding coefficient. Models within a Δ_BIC_ and Δ_AICc_ value of less than 2.77 were considered equivalent (see [73], Box 2, for discussion of the strength of evidence represented by differences of information criteria). Δ_BIC_ was chosen as our primary evidence function due to ease of calculation and to be most conservative about detection of the presence of inbreeding depression [74]. All models were sampled with WinBUGS version 1.4.3 [60], using the dclone package [57] in R version 2.15.0 [64].

The length of each chain for models investigating the association between the individual inbreeding coefficient and a given fitness surrogate was set based on visual inspection of the trace plots and to ensure the value of the $\hat{R}$ statistic [65], as calculated with the coda package [66], was below 1.1 in the most saturated model for all parameters. For data cloning, each model was fit on 1, 5, 10, 20, and 40 total copies of the data, with individual inbreeding coefficients modeled with beta distributions parameterized using the modal values for the number of loci with alleles identical by decent in an individual, as described previously in the methods. The convergence of the data-cloning algorithm was determined using the statistics outlined in [54], as implemented in the dclone package [57]. These statistics were (1) the largest eigenvalue of the posterior variance covariance matrix, which checks the degeneracy of the posterior distributions; (2) the mean squared error; and (3) a correlation-like fit statistic, which, together, check the sufficiency of the normal approximation [57]. We also plotted the logarithm of the posterior variances scaled based on the posterior variance observed using the original data to ensure that they had a linear decrease with the logarithm of the number of clones, and that ultimate scaled variance was below 0.05 [57]. Models whose parameters failed to converge to a multivariate normal distribution based on 40 total copies of the data were removed from the subsequent analysis.

Estimates of regression parameters were calculated as the mean of the posterior distribution of the model fit on 40 total copies of the data, with 95% Wald’s confidence intervals calculated in the dclone package [57]. We also developed confidence bands and prediction bands for each model.

Confidence bands were developed by taking 10,000 random draws from a multivariate normal distribution with a vector of means and a variance covariance matrix calculated from all MCMC chains post convergence for the model fit on 40 total copies of the data. Confidence bands were calculated as the 2.5 and 97.5% quantiles of the distribution of estimated individual fitness trait values at a given inbreeding coefficient.

Prediction bands for these individual regressions were developed by calculating the variance–covariance matrix of regression coefficients from the model fit on 40 total copies of the data. This matrix and the mean coefficient values were then used to parameterize a new WinBUGS model, in which posterior draws of individual based instantaneous population growth rate, individual maximum ages, age at maturity, or wait time between reproductive events were saved for each of 1000 equally spaced inbreeding coefficient values on the interval (0, 0.25). This interval was chosen to span the data-cloned estimates of observed individual inbreeding coefficients in the sample. Prediction bands were calculated as the 2.5 and 97.5% quantiles of the draws of individual fitness trait values at a given inbreeding coefficient from a single chain run for 10,000 iterations, saving every 5th iteration after discarding the first 5000 samples. For more detail on the construction of prediction bands from data-cloned models, see [57].

#### 2.5.1. Lifetime Reproductive Success

We modeled the following:(1)$$r_{i}\sim N\left(\mu_{i},\sigma_{r}^{2}\right)$$
where this normal distribution was truncated below zero and censored above 0.3819509. The distribution was truncated at zero because, by including only reproductive individuals in this analysis, the minimum individual-based instantaneous population growth rate was zero (a single pup produced). In addition, the distribution was censored above 0.3819509 because this was the maximum observable value across the first 19 years of an individual’s life (corresponding to a seal maturing at four years of age and producing a single pup every year until age 19). A normal distribution was selected because an instantaneous population growth rate is commonly assumed to be normally distributed [75], an assumption which has been found to hold for three Australian mammal populations [76].

The richest model fit was as follows:(2)$$\mu_{i}=\beta_{0}+\beta_{1}{*F_{i}+\beta }_{2}*{Col}_{i}+\beta_{3}*F_{i}*{Col}_{i}+\gamma$$
where ${Col}_{i}$ was an indicator variable whose value was one if an individual was born in a Dellbridge Island colony and zero otherwise. The k regression parameters, k = 0, …, 3, were estimated using priors of $\beta_{k}\sim N\left(0,\sigma^{2}=10000\right)$, with the intercept parameter, $\beta_{0}$, truncated between 0 and 0.3819509, using the truncated normal distribution in WinBUGS [77]. These prior distributions were selected to be flat over a much larger region than the range of expected parameter values [78]. We modeled $\sigma_{r}\sim half\text{-}Cauchy\left(0,0.013\right)$ and $\gamma \sim N(0,\sigma_{\gamma }^{2})$ where $\sigma_{\gamma }\sim half\text{-}Cauchy\left(0,0.004\right)$. The scale parameter for these half-Cauchy prior distributions was set to be four times larger than the observed standard deviations in the data [79]. For these models, the individual inbreeding coefficient was modeled as $F_{i}\sim Beta(A_{i},B_{i})$. The values of $A_{i}$ and $B_{i}$ were the modal values of the shape and scale parameters of the posterior distribution of individual inbreeding coefficients calculated in the first step of the analysis. Individual-based instantaneous population growth rate models were run on three independent chains (starting values drawn from $\beta_{0}\sim N\left(0.15,\sigma^{2}=0.0001\right)$, $\beta_{c(1,2,3)}\sim N\left(0,\sigma^{2}=0.01\right)$, $F_{i}\sim Unif\left(0.01,0.1\right)$, $\sigma_{r}=\left|\xi \right|/\sqrt{\chi }$, $\xi \sim N\left(0.1,\sigma^{2}=0.000225\right)$, $\chi \sim N\left(0.3,\sigma^{2}=0.36\right)$, $\sigma_{\gamma }=\left|\zeta \right|/\sqrt{\tau }$, $\zeta \sim N\left(0.002,\sigma^{2}=0.000025\right)$, and $\tau \sim N\left(0.4,\sigma^{2}=0.36\right)$) for 20,000 iterations, saving every 5th iteration after discarding the first 15,000 samples.

#### 2.5.2. Survival across the Study

The probability of and individual’s survival, *s_i_*, was assumed constant across all ages of mature seals [36]. This decision was based on the best evidence available at the time of analysis. Subsequent work [80] found that the survival of reproductive females varies with age; however, most of these declines in survival occur after age 20, an age that is beyond what is being considered in this paper. We modeled age of death as
(3)$${death}_{i}\sim Geom\left(s_{i}\right)$$
and censored it between 4 and 19 to account for the fact that seals were not commonly observed between their birth year and their age at primiparity, the minimum age of which is 4 [37], and the minimum number of years for which all individuals had been tracked was 19. The richest model that was fit to the data was as follows:(4)$$\log\left(\frac{\left(1-s_{i}\right)}{s_{i}}\right)=\beta_{0}+{(\beta }_{1}+\theta_{Y}){*F_{i}+\beta }_{2}*{Col}_{i}+\beta_{3}*F_{i}*{Col}_{i}+\gamma_{Y}$$
where ${Col}_{i}$ and $F_{i}$ were identical to the regression models fit to *r_i_*. The parameters $\theta_{Y}$ and $\gamma_{Y}$ were group level variables for the year (*Y* = 1980, …, 1992) in which an individual was born. The k regression parameters, k = 0, …, 3, were modeled as $\beta_{k}\sim N\left(0,10000\right)$; $\theta_{Y}\sim N(0,\sigma_{\theta }^{2})$**,** where $\sigma_{\theta }\sim Unif\left(0,100\right)$; and $\gamma_{Y}\sim N(0,\sigma_{\gamma }^{2})$**,** where $\sigma_{\gamma }\sim Unif\left(0,100\right)$. Age-of-death models were run on three independent chains (starting values drawn from $\beta_{k}\sim N\left(0,\sigma^{2}=1\right)$, $\theta_{Y}\sim N\left(0,\sigma^{2}=1\right)$, $\gamma_{Y}\sim N\left(0,\sigma^{2}=1\right)$, $\sigma_{\theta }\sim Unif\left(0,10\right)$, and $\sigma_{\gamma }\sim Unif\left(0,10\right)$) for 20,000 iterations, saving every 5th iteration after discarding the first 15,000 samples.

#### 2.5.3. Age at First Reproduction

The age of an individual’s first reproduction was modeled as follows:(5)$${rep}_{i}\sim NBin\left(4,p_{i}\right)$$

We needed to observe four events of yearly survival because those correspond to the minimum age at maturity in the population [37]. The minimum age at maturity was subtracted from all responses prior to model fit. The richest model was the following:(6)$$\log\left(\frac{4*(1-p_{i})}{p_{i}}\right)=\beta_{0}+{(\beta }_{1}+\theta_{Y}){*F_{i}+\beta }_{2}*{Col}_{i}+\beta_{3}*F_{i}*{Col}_{i}+\gamma_{Y}$$
where ${Col}_{i}$ and $F_{i}$ were identical to the regression models fit to *r_i_*, and $\theta_{Y}$ and $\gamma_{Y}$ were group level parameters given the year of an individual’s birth (*Y* = 1982, …, 1992). The k regression parameters, k = 0, …, 3, were modeled as $\beta_{k}\sim N\left(0,\sigma^{2}=10000\right)$; $\theta_{Y}\sim N(0,\sigma_{\theta }^{2})$, where $\sigma_{\theta }\sim Unif\left(0,100\right)$; and $\gamma_{Y}\sim N(0,\sigma_{\gamma }^{2})$**,** where $\sigma_{\gamma }\sim Unif\left(0,100\right)$. Age-at-first-reproduction models were run on three independent chains (starting values drawn from $\beta_{k}\sim N\left(0,\sigma^{2}=1\right)$, $\theta_{Y}\sim N\left(0,\sigma^{2}=1\right)$, $\gamma_{Y}\sim N\left(0,\sigma^{2}=1\right)$, $\sigma_{\theta }\sim Unif\left(0,10\right)$, and $\sigma_{\gamma }\sim Unif\left(0,10\right)$) for 10,000 iterations, saving every 5th iteration after discarding the first 5000 samples.

#### 2.5.4. Frequency of Reproduction

Finally, the wait time between pups was modeled as
(7)$${wait}_{i}\sim Geom\left(\pi_{i}\right)$$
after subtracting a single year from each individual’s response. The probability that a seal reproduced in a given year was assumed constant for all pups, except if it was following the first pup [37], which was based on the best information available at the time but a simplification of the latest findings for females in the population [80]. The richest model fit was as follows:(8)$$\log\left(\frac{\left(1-\pi_{i}\right)}{\pi_{i}}\right)=\beta_{0}+{(\beta }_{1}+\theta_{Y}){*F_{i}+\beta }_{2}*{Col}_{i}+\beta_{3}*F_{i}*{Col}_{i}+\beta_{4}*{First}_{i}+\gamma_{Y}$$
where ${Col}_{i}$ and $F_{i}$ were identical to the regression models fit to *r_i_*, and $\theta_{Y}$ and $\gamma_{Y}$ were group level parameters given the year of an individual’s birth (*Y* = 1982, …, 1992). The variable ${First}_{i}$ had a value of one if the wait time was following an individual’s first pup, and zero if not. The k regression parameters, k = 0, …, 4, were modeled as $\beta_{k}\sim N\left(0,10000\right)$; $\theta_{Y}\sim N(0,\sigma_{\theta }^{2})$, where $\sigma_{\theta }\sim Unif\left(0,100\right)$; and $\gamma_{Y}\sim N(0,\sigma_{\gamma }^{2})$, where $\sigma_{\gamma }\sim Unif\left(0,100\right)$. Individual wait-time models were run on three independent chains (starting values drawn from $\beta_{k}\sim N\left(0,\sigma^{2}=1\right)$, $\theta_{Y}\sim N\left(0,\sigma^{2}=1\right)$, $\gamma_{Y}\sim N\left(0,\sigma^{2}=1\right)$, $\sigma_{\theta }\sim Unif\left(0,10\right)$, and $\sigma_{\gamma }\sim Unif\left(0,10\right)$) for 20,000 iterations, saving every 5th iteration after discarding the first 15,000 samples.

## 3. Results

The expected heterozygosity was estimated as 0.716 (95% confidence interval from 0.668 to 0.764) in this population of seals, with an average of 10.2 alleles per genotyped locus. Expected heterozygosities at our newly developed microsatellite markers were within the range observed in previously described loci (Lc-28 = 0.297, and Lc-6 = 0.888; Table 1), although these loci, in general, had fewer observed alleles (average observed in previously described loci = 11.2, Table 1). In addition, the expected heterozygosity was consistent with observed levels in non-exploited placental mammals (0.677 ± 0.01 SE; [81]), despite harvest occurring in this population of Weddell seals in the 1950s and 1960s [82]. We did not detect linkage disequilibrium between loci following a Benjamini–Hochberg false-discovery-rate correction ([83]; smallest *p*-value = 0.0012, and adjusted α = 0.0001). Of the 29 amplified loci, 1 had an excess of homozygotes (Lew-001677, *p*-value = 0.00, and F_IS_ = 0.1076) based on estimated F_IS_ values [84]. This locus was retained in the final analysis because the estimated F_IS_ value was within the range observed across loci (−0.1549 to 0.1078). However, an excess of homozygotes could indicate that this newly described locus contains a null allele. With no observed null homozygotes in the sample, we estimated that the frequency of a potential null allele would be 0.0283, using the method of [85]. Because we sampled only a single population, we were unable to examine whether there was a consistent pattern of an excess of homozygotes across populations. Future studies genotyping this locus should therefore continue to investigate the potential existence of a null allele.

Across the 29 microsatellite loci, individual homozygosity was correlated with estimated inbreeding coefficients (ρ = 0.67, Figure 1), but identity disequilibrium ($g_{2}$= −0.001) was not greater than 0 (*p*-value = 0.76, and 95% confidence interval from −0.003 to 0.001).

Based on BIC values, the regression model estimating the individual-based instantaneous population growth rate as a function of individual inbreeding coefficient was equivalent to the null model (Table 2). Based on this sample of 154 individuals, we estimated that, for every increase in the individual inbreeding coefficient of 0.0625 (the equivalent of a first cousin mating), there was an associated change in the individual-based instantaneous population growth rate of approximately −0.0066 (95% confidence interval from −0.0154 to 0.0022). This estimated decrease corresponds to a lifetime reproductive success for an individual that was the product of a first-cousin mating that is 96.6% as large as a non-inbred individual (95% confidence interval from 91.5% to 101.1%). While neither BIC nor AICc values indicated that including individual inbreeding coefficients in the model led to a better fit (Table 2), and the 95% confidence interval around the regression parameter for individual inbreeding coefficient included zero, the maximum likelihood estimate for inbreeding suggests that inbreeding may reduce the lifetime reproductive success of individual Weddell seals in this population to some small degree (Figure 2). The two information criteria differ as to whether the no inbreeding depression model or an inbreeding depression model has the lowest criterion value, but both agree that the models are not distinguishable. The scientific conclusions are the same for both criteria.

We estimated that, for an increase in the individual inbreeding coefficient of 0.0625, there was approximately an associated 1.02-times increase in the mean maximum observed age (95% confidence interval from 0.86 to 1.22 times). We also estimated that an increase in the individual inbreeding coefficient of 0.0625 corresponded approximately to a 1.04-times increase in the mean age at maturity (95% confidence interval from 0.92 to 1.18 times). Finally, after accounting for whether a given pup was the mother’s first, there was an estimated increase of approximately 1.06 times in the wait time between pups for every increase in the individual inbreeding coefficient of 0.0625 (95% confidence interval from 0.94 to 1.20 times). The average estimated wait time following the first pup in this population was approximately 2.08-times longer than for subsequent pups across all inbreeding coefficient values (95% confidence interval from 1.59 to 2.73 times).

## 4. Discussion

We used molecular data to estimate the association between inbreeding and lifetime reproductive success of mature female Weddell seals in Erebus Bay, Antarctica. The use of a two-step maximum likelihood analysis allowed us to incorporate uncertainty in estimating individual inbreeding coefficients from multilocus genotypes into our assessment of inbreeding depression in this population.

Using a classical hypothesis testing analysis, we were unable to reject the null hypothesis of no inbreeding depression. This is a null result with very ambiguous interpretation because, in the hypothesis-testing paradigm, *absence of evidence cannot be taken as evidence of absence*. As a consequence, the publication of this work was tabled for almost a decade. A reanalysis in an evidential framework was able to present this work in a much clearer and more useful fashion: the model of no inbreeding depression is statistically indistinguishable here from a model of very slight depression, but both are distinguishable from models with even modest levels of inbreeding depression. What we can say is that there is evidence for, at most, low levels of inbreeding depression in this population. For the conservation biology of Weddell seals, this is very good news—a statistical null hypothesis is not necessarily a biologically null hypothesis.

The data suggest that if inbreeding affects lifetime reproductive success in this population of seals, the effect is very minor. The estimated cost of inbreeding (δ *sensu* [86]) in lifetime reproductive success for mature individuals with an inbreeding coefficient of 0.25 (equivalent to a full sibling mating) was lower (0.137, 95% confidence interval from 0.032 to 0.260) than the average cost of inbreeding for mortality in wild (2.155 after standardizing for *F* = 0.25; [10]) or captive mammal populations (0.33; [9]). It is important to note that these costs of inbreeding are calculated at an inbreeding coefficient beyond any of the median values observed in our sample. In addition, the conclusion that the effect of inbreeding depression is small is supported by the fact that the estimated 5.5% reduction in lifetime reproductive success (95% confidence interval from 4.7 to 6.5%) of mature female seals with an inbreeding coefficient of 0.1 is at the low end of the expected range of inbreeding depression based on a rule of thumb derived from agricultural systems ([87] reported by [12]). Therefore, this low cost of inbreeding in lifetime reproductive success indicates that inbreeding depression does not play as large a role in determining the lifetime reproductive success of mature female Weddell seals as it does in shaping fitness parameters in other mammalian or in agricultural populations.

Although the estimated value of inbreeding depression in the lifetime reproductive success of female Weddell seals in this population from Erebus Bay was low, inbreeding was detected, thus reducing juvenile survival of Weddell seals, in an isolated population on White Island that was founded by individuals from Erebus Bay [42]. Therefore, we may not have detected a strong signature of inbreeding depression because juvenile survival was not included in our analysis. In estimating the lifetime reproductive success of female seals in this population, we analyzed only mature individuals. Seals that died before reproducing were not included in this analysis. It is possible that inbreeding depression in Weddell seals may manifest itself early in the seals’ life and have little effect upon adults that reach breeding age. Because our study did not cover the entire lifespan of individuals, it is also possible that, in addition to juvenile survival, inbreeding depression may affect senescing individuals, but this should have very little effect on *r_i_* [88,89]. As an example, the *r_i_* of a female who survived to 29 years and pupped every other year, starting at age 8, is only about 1% greater than the *r_i_* for the same life table truncated at age 19.

By tracking individuals across 19 years, we captured only a single year of the senescent period for body mass in female Weddell seals [33] and missed the bulk of senescent declines in annual survival rate [80], and, as such, we may not have been able to detect the effects of senescence in this population. Therefore, as this population is the focus of a long-term study, the effects of inbreeding on the senescence of seals should be revisited once all genotyped individuals have completed their life cycle.

In this study, we were unable to detect previously identified associations between either the colony of birth or the year of birth, and any of the life history parameters we investigated. This fact raises the possibility that our sample was not representative of the larger population. However, the similarity between our estimated probability of survival, age at maturity, and reproductive interval and previous findings in this population indicates that this is not the case. For example, we estimated the annual probability of survival for a non-inbred mature individual to be 0.945 (95% confidence interval from 0.933 to 0.954), which compares well with previous reported probabilities of survival for mature individuals (0.94 [36]; 0.905–0.942 [39]; 0.93–0.94 [37,80]). Our estimated mean age at first reproduction for non-inbred seals of 7.39 years (95% confidence interval from 6.93 to 7.92 years) matched well with previously reported estimates in this population (7.67 years [36]; 7.55–8.45 years [37]). We estimated the probability of reproduction in the year following an individual’s first pup to be 0.548 (95% confidence interval from 0.442 to 0.649), which was close to the estimated probability (0.46) of [37]. We also estimated the probability that an experienced breeder produced a pup the following year to be 0.716 (95% confidence interval from 0.684 to 0.746), which was slightly higher than was previously reported (0.67, SD = 0.09) by [37] and lower than what was reported for females 8 to 19 years of age by [80].

While small effect sizes are typical of other studies of heterozygosity fitness correlations [26], the imprecise measure of inbreeding coefficients with only a small number of markers could lead to an underestimate of inbreeding depression. In addition, the lack of identity disequilibrium identified in this sample of Weddell seals could be caused by the number of genotyped loci. For example, Ref. [90] found that nearly half of their simulations with a similar numbers of microsatellite loci failed to detect statistically significant identity disequilibrium even when statistically significant heterozygosity fitness correlations existed. Therefore, the association between inbreeding and lifetime reproductive success could strengthen if investigated with larger numbers of markers that are now available through next-generation sequencing.

The two-step model presented in this study provides a method for incorporating some of the uncertainty inherent in estimating individual inbreeding coefficients from multilocus genotypes into estimates of inbreeding depression. This analysis method allowed us to provide an appropriate context (i.e., the maximum plausible amount of inbreeding depression in this population was modest) for the observed level of inbreeding depression in this non-pedigreed population of Weddell seals.

## 5. Conclusions

While this study is an example of the file-drawer effect, it also highlights the benefit of taking an evidentiary approach to comparing competing models as opposed to focusing on statistical hypothesis testing alone. We were unable to reject the null hypothesis that inbreeding depression does not reduce the lifetime reproductive success of Weddell seals in this population. A strict focus on the failure to reject the null hypothesis would miss the biological implications identified in this study. We were able to weigh evidence for the competing hypotheses and estimate the impact of inbreeding on individual lifetime reproductive success in this population by selecting between competing biological models. With this approach, we found evidence that is consistent with, at most, a slight decrease in lifetime reproductive success at values of inbreeding detected in this population.

Weighing evidence for competing biological models can allow managers to assess the potential impacts of alternate approaches and determine how to best use limited conservation resources more accurately. For example, our study found only a minor influence of inbreeding depression on individual lifetime reproductive success, indicating that inbreeding is not currently a primary concern to the conservation of this population of Weddell seals.

## Figures and Tables

**Figure 1 entropy-25-00403-f001:**
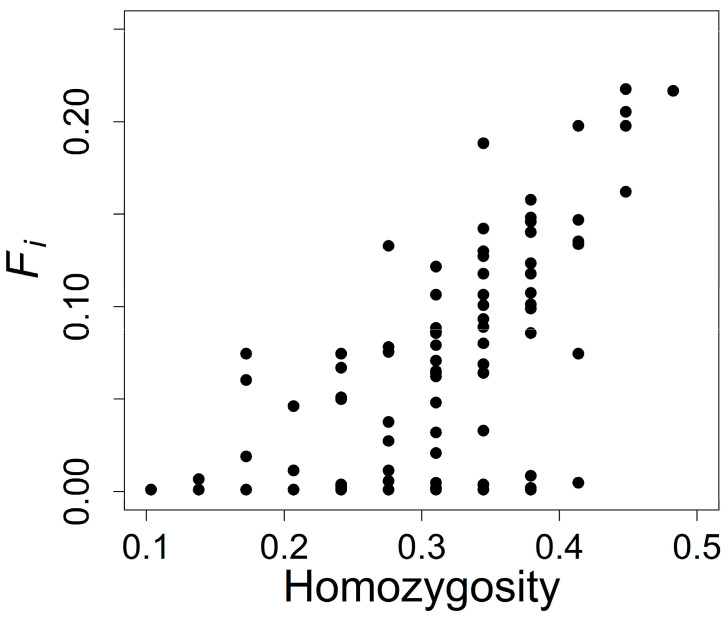
Individual homozygosity at 29 microsatellite loci and estimated individual inbreeding coefficients for 154 Weddell seals.

**Figure 2 entropy-25-00403-f002:**
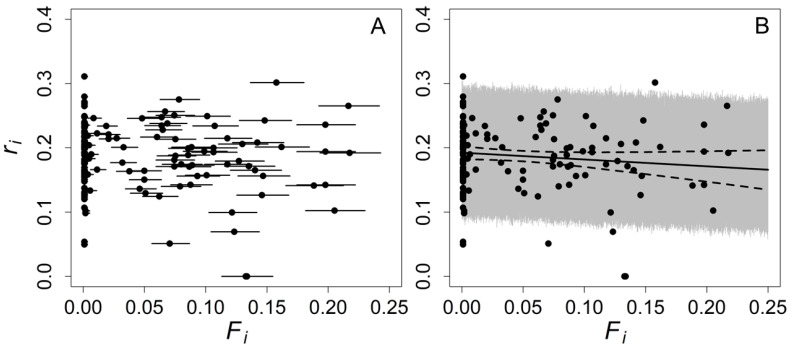
Lifetime reproductive success, measured as an individual-based instantaneous population growth rate, given an individual seal’s inbreeding coefficients. Panel (**A**) shows the point estimates and the 2.5% and 97.5% quantiles of the distributions of individual inbreeding coefficients estimated with the final data-clone model fit in the first step of the analysis. Panel (**B**) only shows point estimates for individual inbreeding coefficients for clarity. The solid line in Panel (**B**) represents the estimated association between individual inbreeding coefficient and the lifetime reproductive success of a seal based on a truncated-normal-regression model fit to censored data, with the 95% prediction band for this analysis denoted by the gray shading and the 95% confidence band for the regression denoted by the dotted lines.

**Table 1 entropy-25-00403-t001:** Primer sequences and marker information for the 12 novel microsatellite loci used in this study.

Locus	Primer Sequences 5′-3′		Repeat Motif	Size (bp)	Number of Alleles	H_E_
*Lew-001339*	F	GCACTCCAGTTTCCTTGGAC	(AC)_14_	106–135	10	0.631
	R	AGGAGCTTAGTAGGCAATCC				
*Lew-001677*	F	ACAAGGGATTCTTAGGGAACTG	(AC)_15_	214–231	15	0.799
	R	TCCAGTGGTAATAACTTGCAAAC				
*Lew-001845*	F	TGTAACCTCAAGGGTCCCAC	(TG)_12_	147–153	4	0.519
	R	GCGTCTGGAGTGTGGAATTG				
*Lew-001859*	F	TCTCCCTGTTCATTAGATCCTG	(AC)_13_	98–114	8	0.622
	R	GAGCCAACTTGCATTGTGTTC				
*Lew-001873*	F	TGTTTCCGGTTGGGCTATTC	(TG)_14_	133–155	8	0.624
	R	ACGATAGATTGGGCCTTGTC				
*Lew-002658*	F	ATTCTCAGACCTCAGGGAGC	(GT)_16_	122–146	8	0.709
	R	CATCCTGAGTTTGGCCTTGG				
*Lew-002762*	F	TGTTCCATCTCCTGCCACTC	(AC)_13_	164–185	9	0.759
	R	ATCTGGGGAAAGGTGGGTTC				
*Lew-004467*	F	TGCACAGTATAAAACAGGATAGAGG	(AC)_14_	90–111	11	0.788
	R	CCAGAGAGAGCCTGTGTACG				
*Lew-005761*	F	AGAGAGGGTCATTAGAGACAGC	(AC)_13_	158–174	8	0.782
	R	ATGACTCTTCATGGGCGTGC				
*Lew-006174*	F	TGGTGAACTCAACAAGGGAAAG	(AC)_15_	101–112	7	0.738
	R	TGTATTGCTCAGCCCAACTC				
*Lew-006657*	F	GCATGCTGGGTCATGAGTG	(GT)_13_	82–103	11	0.785
	R	GCCCCACGATGTTACTAAGTTG				
*Lew-007425*	F	AAGTTTTATGTGGGCATCCG	(GT)_12_	96–106	5	0.560
	R	GCCGTTCACATTTCTGCCTC				

**Table 2 entropy-25-00403-t002:** Candidate regression models for each of the individual fitness surrogates ranked based on their relationship to the model with the minimum BIC value, with relative AICc values also reported. Only those models that converged are shown. Explanatory variables included individual inbreeding coefficient (*INB*), birth location (*COL*), and a variable in the mean wait time between pupping intervals’ models, indicating whether the interval was following an individual seal’s first pup (*FIRST*).

Model Structure		K	Δ_BIC_	Δ_AICc_
*r_i_*				
	$\beta_{0}$	3	0.00	2.63
	$\beta_{0}+\beta_{1}INB$	4	0.32	0.00
	$\beta_{0}+\beta_{1}INB+\beta_{2}COL$	5	3.34	0.09
	$\beta_{0}+\beta_{1}INB+\beta_{2}{COL+\beta }_{3}INB*COL$	2	8.39	2.24
Minimum age				
	$\beta_{0}$	1	0.00	0.00
	$\beta_{0}+\beta_{1}INB$	2	4.53	1.55
	$\beta_{0}+\beta_{1}INB+\beta_{2}COL$	3	9.44	3.5
	$\beta_{0}+\beta_{1}INB+\beta_{2}{COL+\beta }_{3}INB*COL$	4	14.34	5.47
Age at maturity				
	$\beta_{0}$	1	0.00	0.00
	$\beta_{0}+\beta_{1}INB$	2	4.55	1.57
	$\beta_{0}+\beta_{1}INB+\beta_{2}COL$	3	9.51	3.57
	$\beta_{0}+\beta_{1}INB+\beta_{2}{COL+\beta }_{3}INB*COL$	4	14.55	5.68
Pupping interval				
	$\beta_{0}$ + $\beta_{4}FIRST$	1	0.00	0.00
	$\beta_{0}+\beta_{1}INB+\beta_{4}FIRST$	2	3.99	1.04
	$\beta_{0}+\beta_{1}INB+\beta_{2}COL+\beta_{4}FIRST$	3	7.96	2.07
	$\begin{array}{c} \beta_{0}+\beta_{1}INB+\beta_{2}COL+\beta_{3}INB*COL\\ +\beta_{4}FIRST \end{array}$	4	11.61	2.83

## Data Availability

Data used in the analysis are available in the Appendix A.

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
