# Peer review of "Evidence of an Absence of Inbreeding Depression in a Wild Population of Weddell Seals (Leptonychotes weddellii)"

_entropy, 2023, doi:10.3390/e25030403_

Round 1
Reviewer 1 Report
The manuscript, which reports a slight correlation between inbreeding and reproductive success, is well-written and can provides a clear analytical framework for for future studies with similar aims.
I don't see major flaws in the design of analyses. Some minor comments are attached to the pdf file. Overall, the manuscript can be more succinct by citing or describing equations and parameters in the Methods section and reducing come repetitive content in the Results section.
I value the framework of the study more than the results, therefore, I'd suggest authors to attach annotated scripts of analyses in the supplementary material or make them available in open online repositories.

Author Response
We thank the reviewer for their helpful comments on improving the clarity of the manuscript and future utility of the work. In addition to addressing their comments on the text of the manuscript, we have added a supplemental file with the WinBUGS models used in the analysis so they are available to other researchers. Below is a table identifying changes to the text of the manuscript in response to the reviewer’s comments.
|
Reviewer Comment |
Response |
|
Line 20: This sentence sounds ambiguous, I'd suggest author to make it clearer. |
This sentence was restructured to improve clarity. |
|
Line 46: Cite any examples? |
We added a reference for this statement and additional discussion of the issue on lines 46-57. |
|
Line 115: typo |
mere was corrected to were |
|
Lines 178-205: These equations are not necessary as the authors already cited Vogl et al. |
The equations were removed from the manuscript for clarity. |
|
Lines 323-336: I think these settings of parameters can be mentioned as text and not necessarily to be presented as formula here. |
The formulas describing model settings and parameters were collapsed into the text. |
|
Lines 351-367: I think these settings of parameters can be mentioned as text and not necessarily to be presented as formula here. |
The formulas describing model settings and parameters were collapsed into the text. |
|
Lines 374-389: I think these settings of parameters can be mentioned as text and not necessarily to be presented as formula here. |
The formulas describing model settings and parameters were collapsed into the text. |
|
Lines 396-414: I think these settings of parameters can be mentioned as text and not necessarily to be presented as formula here. |
The formulas describing model settings and parameters were collapsed into the text. |
Reviewer 2 Report
Powell et al. present a well written manuscript with genetic data that were outdated by new sequencing methods, but are nevertheless interesting. I appreciate that an analysis that gave not the preferable results is aimed to publish and that errors in the x-variable in a regression analysis are taken into account. Without taking that error into account regressions are generally biased downward. Nevertheless, in this study there was no significant difference in the fit of the models with and without inbreeding coefficients. However, this fact is not reflected by the title and conclusion in the abstract, which surrogate that there is clear inbreeding depression (see comments below).
General comments:
The statement of evidence for inbreeding depression should be more attenuated in the title and the abstract. Given that the models with and without inbreeding are not different based on AIC there is no clear evidence to inbreeding depression nor that it is not occurring in the population. Maybe use “suggestive” evidence as it is described in the results part.
It would be informative if the estimated inbreeding coefficients are described more in detail e.g. the correlation of observed homozygosity per individual across all markers with the estimated inbreeding coefficient to be able to compare the estimated inbreeding coefficients with the classical multi locus heterozygosity approach. Moreover to be able to judge how well the inbreeding coefficients from microsatellites represent the true inbreeding coefficients the identity disequilibrium g2, a measure of the covariance in heterozygosity, should be reported, either for the multi locus heterozygosity (e.g. with RMES (David P, Pujol B, Viard F, Castella V, Goudet J. 2007 Reliable selfing rate estimates from imperfect population genetic data. Mol. Ecol. 16, 2474–2487.)) or for the inbreeding coefficients as estimated in the manuscript.
Minor comments:
Materials and Methods:
Line 375: The sentence is unclear: Rephrase, e.g. Four events of yearly survival needed to have been…
Results:
Report about the model fit of 1, 5, 10 and 20 copies of the data is missing.
Discussion:
Note that inbreeding depression might well be larger, because of the imprecise measure of inbreeding coefficients with microsatellites. With larger number of markers of next generation sequencing methods the correlation might strengthen.
Line 488: “associated” seems wrong to me and I would replace it with “association”, but since I am not a native speaker please check.
Conclusion:
574: one “.” too much
Author Response
We thank the reviewer for their helpful comments that improved the manuscript. We agree with the reviewer’s general comments and have addressed them in the manuscript. We incorporated the reviewer comment that “The statement of evidence for inbreeding depression should be more attenuated in the title and the abstract” by changing the title from “Evidence of low inbreeding depression in a wild population of Weddell seals (Leptonychotes weddellii)” to “Evidence of an absence of inbreeding depression in a wild population of Weddell seals (Leptonychotes weddellii).” We also made an identical change in the second to last sentence of the abstract and removed the last sentence of the abstract.
Reviewer 2 also commented that “[i]t would be informative if the estimated inbreeding coefficients are described more in detail in the revised manuscript” and “to be able to judge how well the inbreeding coefficients from microsatellites represent the true inbreeding coefficients the identity disequilibrium g2, a measure of the covariance in heterozygosity, should be reported.” In the revised manuscript we provide readers with the ability to assess the uncertainty in estimated inbreeding coefficients in the analysis by adding a plot of individual homozygosity and estimated inbreeding coefficients, reporting the correlation between these values in the text, and presenting the estimated identity disequilibrium (David et al. 2007) in the text.
Below is a table identifying changes to the text of the manuscript in response to the reviewer’s specific comments.
|
Reviewer Comment |
Response |
|
Line 375: The sentence is unclear: Rephrase, e.g. Four events of yearly survival needed to have been… |
This sentence was restructured to improve clarity. |
|
Report about the model fit of 1, 5, 10 and 20 copies of the data is missing. |
We did not report the model fit for 1, 5, 10, or 20 copies of the data because those were intermediate fits of the data cloning algorithm that were used to assess convergence of model parameters. The use of copies of the data in the data cloning algorithm is clarified in the manuscript (lines 168-175). |
|
Note that inbreeding depression might well be larger, because of the imprecise measure of inbreeding coefficients with microsatellites. With larger number of markers of next generation sequencing methods the correlation might strengthen. |
We added a paragraph to the discussion (lines 543-552) that discusses this uncertainty and the lack of statistically significant identity disequilibrium found in this sample of Weddell seals. |
|
Line 488: “associated” seems wrong to me and I would replace it with “association”, but since I am not a native speaker please check. |
Associated was changed to association |
|
574: one “.” too much |
The additional period was removed. |